# Integrating Ecosystem Services into Land-Use Modeling to Assess the Effects of Future Land-Use Strategies in Northern Ghana

**Hongmi Koo \*, Janina Kleemann and Christine Fürst**

Department of Sustainable Landscape Development, Institute for Geosciences and Geography,
Martin Luther University Halle-Wittenberg, Von-Seckendorff-Platz 4, 06120 Halle (Saale), Germany;
janina.kleemann@geo.uni-halle.de (J.K.); christine.fuerst@geo.uni-halle.de (C.F.)
**\*** Correspondence: hongmi.koo@geo.uni-halle.de; Tel.: +49-(0)-176/24862960

**Abstract:** In West Africa, where the majority of the population relies on natural resources and rain-fed agriculture, regionally adapted agricultural land-use planning is increasingly important to cope with growing demand for land-use products and intensifying climate variability. As an approach to identify effective future land-use strategies, this study applied spatially explicit modeling that addresses the spatial connectivity between the provision of ecosystem services and agricultural land-use systems. Considering that the status of ecosystem services varies with the perception of stakeholders, local knowledge, and characteristics of a case study area, two adjoining districts in northern Ghana were integrated into an assessment process of land-use strategies. Based on agricultural land-management options that were identified together with the local stakeholders, 75 future land-use strategies as combinations of multiple agricultural practices were elaborated. Potential impacts of the developed land-use strategies on ecosystem services and land-use patterns were assessed in a modeling platform that combines Geographic Information System (GIS) and Cellular Automaton (CA) modules. Modeled results were used to identify best land-use strategies that could deliver multiple ecosystem services most effectively. Then, local perception was applied to determine the feasibility of the best land-use strategies in practice. The results presented the different extent of trade-offs and synergies between ecosystem services delivered by future land-use strategies and their different feasibility depending on the district. Apart from the fact that findings were context-specific and scale-dependent, this study revealed that the integration of different local characteristics and local perceptions to spatially explicit ecosystem service assessment is beneficial for determining locally tailored recommendations for future agricultural land-use planning.

**Keywords:** land-use planning; scenario; agriculture; spatially explicit simulation; modeling; stakeholder; participatory assessment

---

## 1. Introduction

The status of ecosystem services (ES) is characterized by consequences of anthropogenic environmental changes and their influence on human well-being and benefits [1–3]. ES assessments, thus, have been considered useful to support land-use and management planning [4]. The potential impacts of land-use decisions on the flow of ES and trade-offs and synergies between different ES help to identify future alternatives for the effective and efficient provision of ES [5–7]. Especially, spatially explicit ES assessments can facilitate the integration of ES in land-use planning by providing information about potential ES mismatches, hotspots, and optimized allocation of land for specific uses [1,4,8,9]. There has been various research that incorporated such ES approaches into land-use planning globally.

Taking case studies of Europe, Asia, and America as an example, the impact of past and current land-use and land-cover changes on the provision of specific ES (e.g., fresh water provision and air quality) was analyzed in order to give an insight into relevant land-use schemes and policies [10,11]. Different future pathways according to social-ecological drivers and their potential implications to the ES status were explored and discussed based on the experts and stakeholders' opinions [12,13]. In addition, changes in multiple ES provision depending on the future land-use and landscape patterns were quantified for identifying optimal future options considering ES trade-offs [14–16]. However, there is still a lack of attempt to apply ES assessments for adapted land-use planning in West Africa, where people are heavily dependent on land-use activities and resultant products and benefits [17,18].

In West Africa, more than 60% of the population is engaged in agriculture and approximately 70% of the land is used for cultivation, which is mainly for rain-fed agriculture [19–21]. High reliance on climate-sensitive farming makes the agricultural systems vulnerable to climate variability and causes high uncertainties about the sustainable supply of food and raw materials [22,23]. In addition, the rapid population growth increases pressures on land-use systems and food security levels [24]. There is an urgent need to reduce risks and to cope with the increasing demand in land use through regionally adapted land-use strategies [25]. However, as one of the poorest regions in the world, its insufficient economic and institutional capacity makes it difficult to properly respond to such situations [19]. The integration of the ES concept in designing agricultural land-use strategies has the potential to support future land-use planning as presented above, but there should be an understanding why such approaches are still not well applied in the West African context. Consequently, there is the need to improve the applicability of the ES concept for land-use planning in West Africa. Firstly, it is necessary to understand how people in the region obtain agricultural land-use-related ES and how they exploit them. In West Africa, land-use products from one type of land are commonly used for various purposes, such as forest for providing food, fodder, construction materials, and fuel [26,27]. Agroforestry as a combination of crops and tree plantations also provides multiple ES such as food, fiber, timber for fuel wood and construction, micro-climate regulation, soil erosion control, pest control, pollination, and carbon sequestration [28,29]. Thus, the multifunctionality of a land-use system needs to be emphasized in ES assessment and future agricultural land-use planning, which provides various benefits to fulfil different economic, social, and ecological requirements by a society [30]. Secondly, the heterogeneous demands on multifunctional land-use systems can be well reflected from perspectives of stakeholders. The consumption patterns of agricultural land-use products of one local community might differ from another community outside the area, and regionally specific distribution of ES influences preferred land-use strategies among stakeholders in the region [31]. Therefore, the participation of stakeholders who own local knowledge regarding agricultural land use in their particular environment is essential for designing future land-use strategies [32,33]. Stakeholders can engage in screening adoptable land-use alternatives by expressing their preferences and perspectives [34,35]. This offers an opportunity to stakeholders as ES beneficiaries to take part in decision-making processes that influence their future lives, and furthermore, their participation can raise the public acceptance of decisions [36,37]. However, participatory approaches might be limited to interpreting the narratives of stakeholders regarding a complex land-use context or uncertain future outcomes [38,39]. Accordingly, assessments that include stakeholder feedback process can be time-demanding and restricted [35,40]. Here, a simulation model can be appropriate to be used for integrating a participatory approach and ES assessments for agricultural land-use planning, which presents potential effects of future land-use decisions through mapping and visualization [41,42]. Visualized simulation results can especially facilitate communication with stakeholders in evaluating process and deriving recommendations [43]. As a simulation approach, spatially explicit modeling in particular is suitable for addressing the spatially variable nature of ES provision linked to the effects of land-use patterns [8,44]. However, existing studies in West Africa are missing either the participatory component or spatially explicit relationships between land use and multiple ES provision. For example, Kleemann et al. [23] used a non-spatially explicit participatory modeling approach for northern Ghana,

but they considered only one ES (food provision) due to increasing complexity in using a bundle of ES. Leh et al. [17] conducted a spatially explicit assessment for the effects of land-use changes on multiple ES provision, but the assessment process was only based on the perspective of scientists rather than actual local perception on ES. Ahmed et al. [45] and Salack et al. [46] identified the interlinkage between land-use systems and climate changes and their impact on food provision, but their results were also not backed-up by local representatives or experts.

In order to investigate the applicability of ES concept for future land-use planning in West Africa, this study suggests a spatially explicit ES modeling approach in combination with stakeholder participation in the agricultural context of Ghana, West Africa. Two districts in northern Ghana were taken as case studies where perspectives of local stakeholders were reflected in an overall assessment process. This included the development of agricultural land-use strategies, simulation conditions of developed land-use strategies, and feedback on the simulated results. Especially, the feedback from the stakeholders on the simulated results was considered as an essential step to present the interaction between local knowledge and land-use modeling. This study is based on the results of previous studies [27,47], which assessed the impacts of various land-use scenarios on the provision of multiple ES in Northern Ghana. They covered the identification of a locally legitimate stakeholder group at district level, the selection of locally relevant ES and indicators, and the development of applicable agricultural land-use scenarios that were applied as management options for elaborating future strategies in this study. Using the previously obtained results and data and newly generated data regarding future land-use strategies and feedback from stakeholders, this study focused on addressing the following research questions:

- How can local perspectives be reflected in identifying the most feasible land-use strategies?
- What kind of synergies and trade-offs appear between ES depending on land-use strategies?
- How do local perspectives and characteristics influence the results on district level?

In addition, methodological and conceptual questions will be discussed:

- What are the advantages and challenges of the applied stakeholder-based ES modeling approach?
- How the application of the ES concept in land-use planning in the West African context can be improved?

Firstly, future land-use strategies were elaborated based on management options for agricultural land, which were expressed in a spatially explicit way. Impacts of the developed land-use strategies on current land-use patterns and ES provision were then assessed in a modeling platform, which combines Geographic Information System (GIS) and Cellular Automaton (CA) modules. According to the simulated results, best land-use strategies were determined that could provide multiple ES most effectively. The feasibility of the best land-use strategies in practice was identified in order to suggest recommendations for future agricultural land-use planning. In the discussion, the strength and weakness of the applied stakeholder-based ES modeling approach and the future directions of using the ES concept for land-use planning in West Africa were discussed.

## 2. Material and Methods

### 2.1. Case Study Area

The study area is located in northern Ghana and includes Bolgatanga Municipal district (hereafter, Bolgatanga) and Bongo district (Figure 1). Bolgatanga covers a total area of 729km$^2$ and, Bongo has a total area of 460km$^2$ [48]. The districts have two seasons—a dry season from October to the beginning of April and a rainy season spanning from April/May to September/beginning of October—with the average annual rainfall ranging between 645mm and 1250mm [49]. Erratic climatic patterns regarding the time of onset, span, and the quantity of rainfall make it difficult to ensure sufficient amounts of water for the various uses. The majority of soil in this area is coarse textured and low in accumulation of organic matter, which is prone to surface runoff by intensified rainfall exceeding the soil infiltration

capacity [50]. Despite of unfavorable conditions for climate-sensitive cultivation, this area still heavily relies on rain-fed small-scale agriculture as do many other West African regions. Approximately 60% of households in Bolgatanga and 96% of households in Bongo are engaged in agriculture, and more than 70% of the land in both districts is used for cultivation [48]. These adjoining districts have similar environmental and land-use conditions. However, each district has an individual political and administrative system due to a decentralization program of Ghana [51]. The decision-making process especially related to agricultural land use is based on agricultural extension services of each district [27]. The land-use pattern of the study area consists of nine land-use types, according to the classification by Forkuor [52].

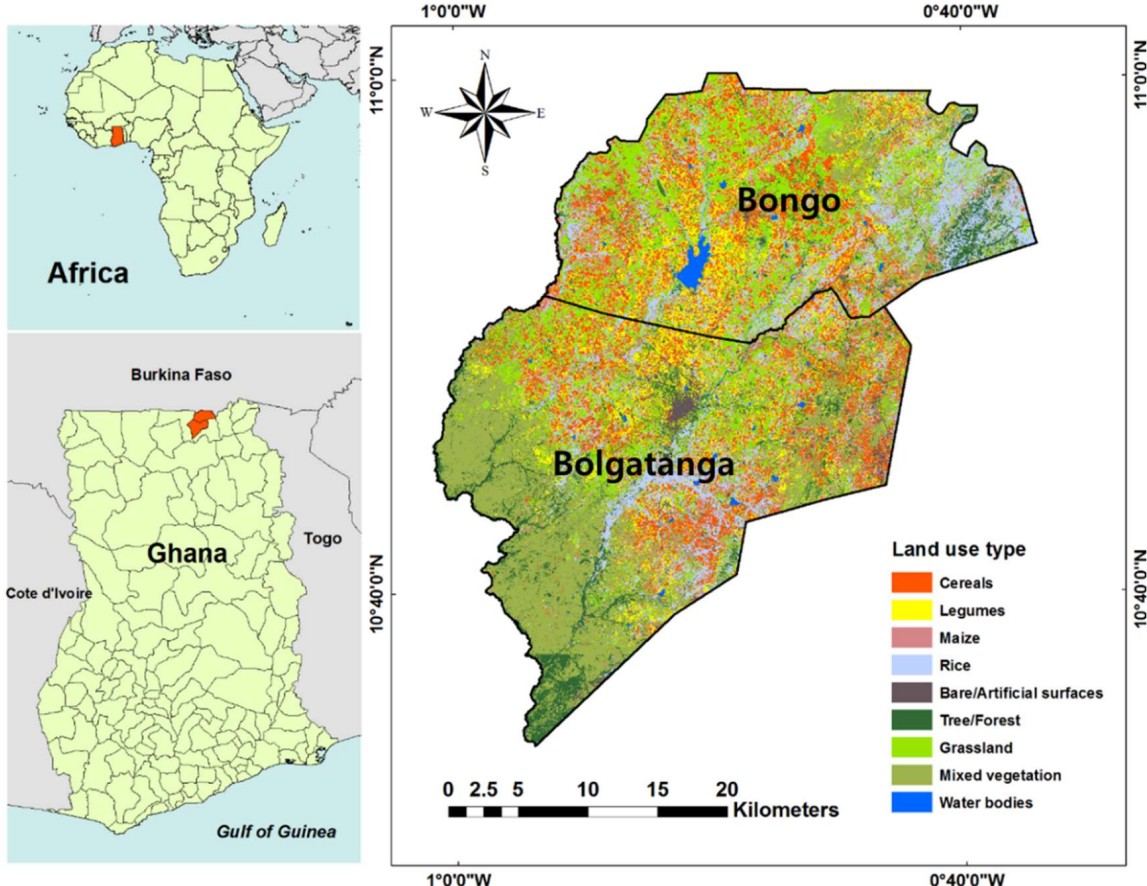

**Figure 1.** Location and land-use patterns of the study area in northern Ghana. The land-use classification is based on RapidEye images of 2013 with resolution of $25 \times 25$ m$^2$. (Forkuor [52]). The description and the areal percentage of each land-use type are presented in Table S1 in the Supplementary Materials.

### 2.2. Database and Selection Processes

This study was based on previous studies in northern Ghana where data have been gathered by Koo et al. [27,47] and integrated in the modeling approach. Here, a short overview of used data and selection processes is provided.

Selection of stakeholders and participatory approach: The stakeholders were selected considering their interest and influence in decision-making on agricultural land use at the district level [27]. Based on interviews with various actors in the agricultural sectors such as farmers, NGOs, and officers of governmental bodies (the Water Resources Commission, the Forestry Commission, the Ministry of Food and Agriculture) and literature [20,53,54], agricultural extension agents (hereafter, extension officers) of the Ministry of Food and Agriculture of Ghana (MOFA) were chosen as stakeholders for this study. Although farmers as direct land users have high interest in farming conditions, their decisions primarily

influence land-use activities at farm level and often indirectly affect agricultural decision-making and policies at district level. On the other hand, extension officers are in charge of several communities with the main duty to give advice in farming, introduce new techniques and policies to farmers, and regularly monitor and report cultivation conditions to the district office of MOFA [55,56]. Since they perform as mediators between farmers and district policy makers, their knowledge and expertise considerably influence land-use practices of farmers and the implementation of agricultural policies and strategies of MOFA [27]. They also highly influence agricultural programs launched by NGOs and governmental authorities as field experts. Extension officers, accordingly, play a decisive role in actual farming decisions and implementation of agricultural policies at district level [47]. All extension officers who are in charge of Bolgatanga (15 officers) and Bongo (11 officers) participated as stakeholders, and their knowledge and opinions were collected through stakeholder surveys. Questionnaires and interviews are common practice in collecting information about the ES perception and valuation [57,58]. Semi-quantitative approaches with questionnaires allow a better comparability of responses than from qualitative approaches using only open questions. In this study, semi-structured and structured surveys with the stakeholders were conducted to generate pertinent information and input for land-use simulation.

Selection of ecosystem services (ES): In this study, ES are defined as human benefits obtained from agricultural land-use activities. Regarding locally relevant ES in the agricultural context, firstly, a preliminary set of ES was identified based on existing ES studies [17,59–66]. The specific ES were then determined from the preliminary list through a semi-structured stakeholder survey [27]. The criteria of the selection were: (1) ES that are perceived to be important for agricultural land use, and (2) ES that can be recognized by their different provisioning levels based on the land-use types (Table 1, [27]). The selected ES were the provision of food, fodder, energy, construction material, marketable product, water, and erosion control. ES values of current land-use types were calculated using the indicators in Table 1. Indicators for the provision of food, fodder, energy, construction materials, and marketable products that are perceived as direct benefits from agricultural land-use activities were developed to reflect local consumptive patterns of varied land-use products (e.g., grains, stalks, branches, and leaves). They were identified through a stakeholder survey [27]. For example, a proportion to be consumed as animal feed out of the entire cereal products inclusive of grains, stalks, and leaves (assumption: 100% of use) was calculated as fodder provision of cereals. With respect to water provision and erosion control as a concomitant and indirect benefits of agricultural activities, proxy indicators from existing studies were used [17,67,68].

**Table 1.** Locally relevant agriculture related ecosystem services and indicators to assess the ecosystem services as identified in Koo et al. [27,47].

| Ecosystem Service | Definition | Indicator |
|---|---|---|
| Food | Benefit of agricultural land-use activities linked with food | Proportion of land-use products consumed as food for households (%) |
| Fodder | Benefit of agricultural land-use activities linked with fodder | Proportion of land-use products consumed as animal feed (%) |
| Energy | Benefit of agricultural land-use activities linked with fuel for household | Proportion of land-use products used for fuel (cooking and heating) (%) |
| Construction material | Benefit of agricultural land-use activities linked with construction materials | Proportion of land-use products used for construction purposes (roofs, pillars) (%) |
| Marketable product | Benefit of agricultural land-use activities linked with economic value | Proportion of land-use products used for selling in the market (%) |
| Water | Surface water yield to flow to water bodies for human direct use | Potential water yields determined through a gap between precipitation and evapotranspiration ($mm\ cell^{-1}yr^{-1}$) |
| Erosion control | Potential to prevent surface run-off | Potential soil erosion level calculated by the RUSLE model ($t\ ha^{-1}yr^{-1}$) |

Selection of land-management options: Future land-use strategies need to be developed considering ES protection, improvement, or trade-off between different ES [69]. Designing land-use strategies based on potential scenarios is useful in terms of the uncertain future development and the investigation of viable actions to implement [70]. In this study, land-use strategies indicate combinations of different agricultural management practices. We assumed that the application of multiple management options as strategies is more effective to enhance various ES than a single management option, since the cumulative positive impact of the management options of each strategy can be expected. The management options used for developing land-use strategies were identified in the previous study [27], and they were associated with crop-intercropping, agroforestry, afforestation, and soil conservation (Table S2 in the Supplementary Materials). In total, 15 agricultural land-management options were selected according to the following criteria: (1) the possibility to mitigate climate change impacts on agricultural areas such as a decrease in land productivity and loss of soil, and (2) the applicability in the local context based on perspectives of stakeholders [27].

Selection of a modeling approach: Since experiments on landscape scale on land-use changes are time-consuming and costly, the simulation of impacts of land-use changes has been widely used [71,72]. The selection of the appropriate land-use model is dependent on characteristics such as non-spatial versus spatial, dynamic versus static, descriptive versus prescriptive, and deductive versus inductive [73,74]. In order to address spatially variable characteristics of ES depending on the modifications of land-use patterns, this study adopted the spatially explicit simulation modeling platform GISCAME that consists of GIS modules and a CA module. This modeling approach allowed us to simulate spatially explicit changes in land-use patterns according to variable scenarios and to visualize their impacts on the ES provision [75].

### 2.3. Development of Future Land-Use Strategies

As explained above, a future land-use strategy is elaborated as a combination of 15 different land-management options [27]. Target land-use types were cereals, maize, legumes, grassland, and mixed vegetation, which have a high likelihood of conversion in the local context. Rice has a low probability of conversion due to its restricted farming conditions and high value in the local market and, therefore, was excluded [76]. In addition, forest cover was excluded because it is mostly influenced by statutory land-use planning of the Town and Country Planning Department and the Forest Commission [77]. All possible combinations of the 15 land-management options were applied to the 5 target land-use types (75 land-use strategies, Figure 2). For instance, future land-use strategy 6 indicates a combination of cereals with crop intercropping (CI), maize with mango agroforestry (MM), legumes with leucaena agroforestry (LL), grassland with afforestation (GA), and mixed vegetation with afforestation (MxA).

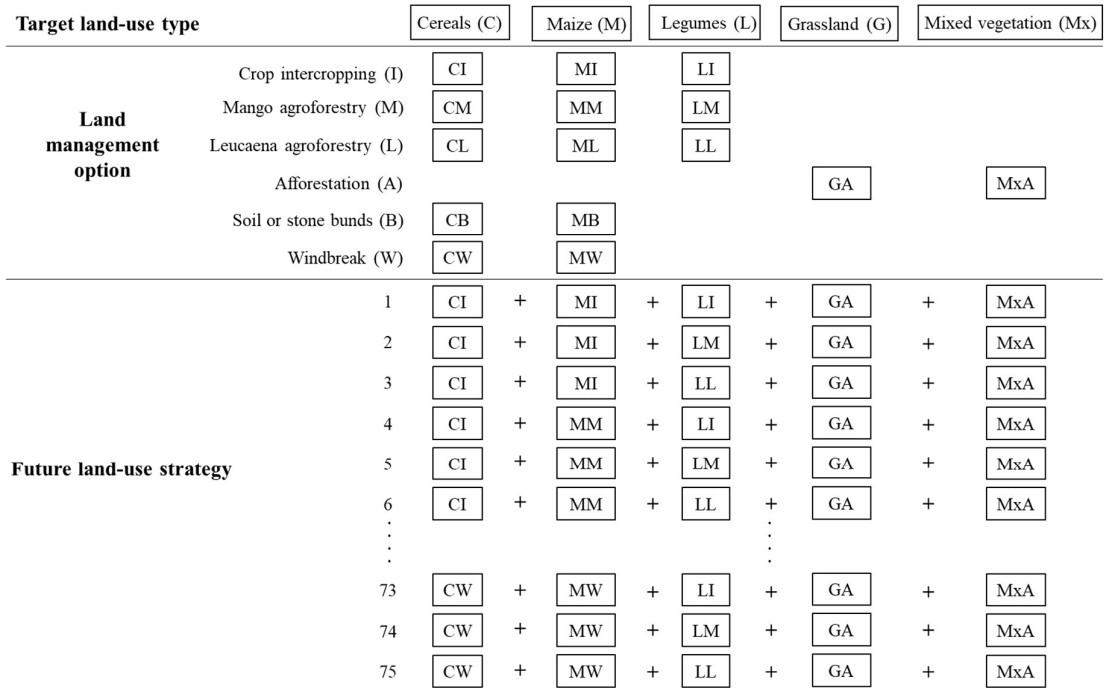

**Figure 2.** Development of future land-use strategies. Land-use strategies are considered as combinations of land-management options applied to target land-use types. Each box indicates which land-use management option applies to which land-use type. For instance, "CI" means a crop intercropping management applied to cereals. The meanings of abbreviated land-use management options are as below: CI: Cereal-dominant intercropping; MI: Maize-dominant intercropping; LI: Legume-dominant intercropping; GA: Grassland afforestation; MxA: Mixed vegetation afforestation; CM: Cereal intercropping with mango; MM: Maize intercropping with mango; LM: Legume intercropping with mango; CL: Cereal intercropping with leucaena; ML: Maize intercropping with leucaena; LL: Legume intercropping with leucaena; CB: Soil or stone bunds on cereals; MB: Soil or stone bunds on maize; CW: Windbreak on cereals; MW: Windbreak on maize.

## 2.4. Assessment Process for Potential Impacts of Land-Use Strategies on Ecosystem Services

The developed future land-use strategies were assessed by the process presented in Figure 3. The impact of a land-use strategy was determined by the combined effect of agricultural land-management options that compose the strategy. At first, the capacity of land-management options for ES provision was identified based on a stakeholder survey (blue boxes in Figure 3, and Section 2.4.1). The ES capacities were expressed in a range from 0 to 100 through standardization. Future land-use patterns influenced by land-use strategies were generated as the next step, in consideration of spatial transition conditions of land-management options in the local context (red boxes in Figure 3, and Section 2.4.2). These two parts were coupled in a modeling platform GISCAME in order to assess ES values at the district level. The best land-use strategies that can potentially provide the highest level of multiple ES (more than three different ES) at the district level were selected based on the simulated results (yellow boxes in Figure 3, and Section 2.4.3). Finally, the feasibility of these best land-use strategies in practice was identified by the local stakeholders in order to derive recommendations for future land-use planning (green boxes in Figure 3, and Section 2.4.3).

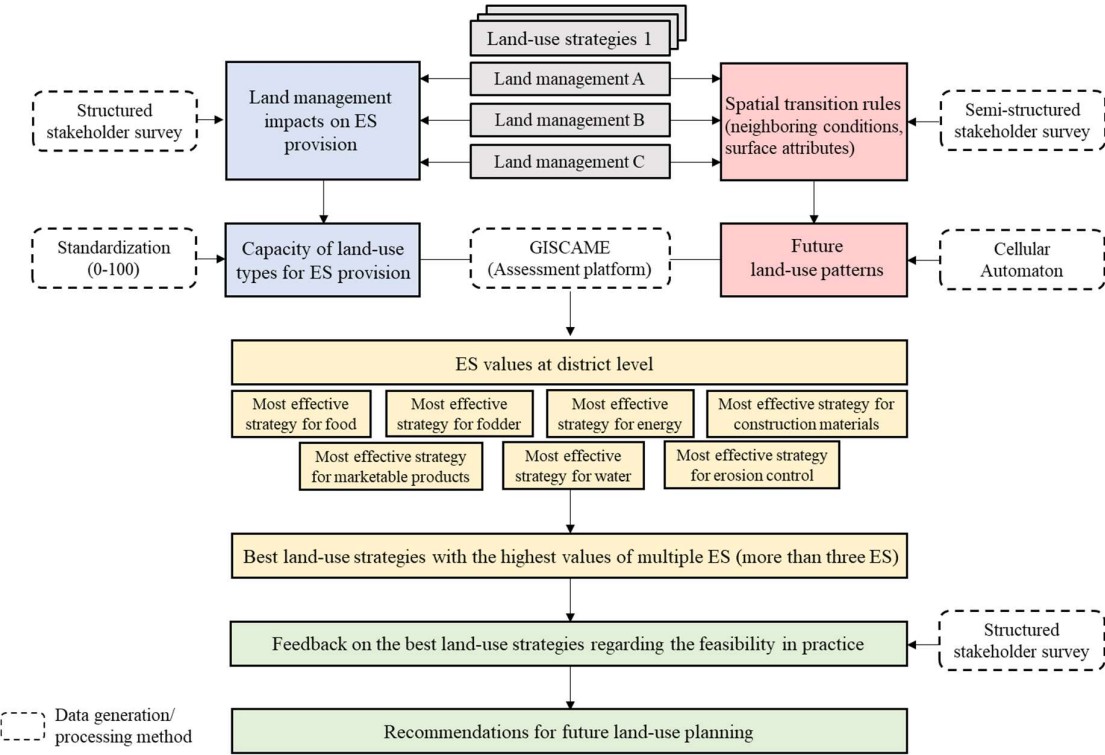

**Figure 3.** Assessment framework of impact of future land-use strategies on the provision of ecosystem services (ES) at district level.

### 2.4.1. Capacity of Land-Use Types to Provide Ecosystem Services

In order to assess the capacity of land-use strategies to supply ES at district level, it is necessary to first identify ES values of land-use types influenced by agricultural land-management options, which comprise the strategies. Here, the ES values of land-use types that were assessed in the previous study [47] were applied, and they were analyzed considering potential trade-offs and synergies between different ES as land-management impacts. For example, mango intercropping with maize as an agroforestry management option can lead to synergies between multiple ES through supplying fruits, firewood, and fence materials, while preventing surface run-off by the root system of mango trees [78,79]. On the other hand, benefits associated with the yield of maize can be reduced because the intercropping practice could have a negative impact on maize growth due to the competition for space, soil nutrients, and water with the mango trees, as a trade-off [80,81]. Such potential impacts were identified based on the experience of the stakeholders. The expected changes were expressed as percentages of increase or decrease compared to the current status of each ES (e.g., 30% potential increase in construction material provision by agroforestry). The final ES values were standardized to a relative scale between 0 (lowest ES provision) and 100 (highest ES provision) in order to compare ES values assigned to land-use types with the same unit [64]. In this assessment, we weighted all selected ES equally. The standardized values were composed of an assessment matrix that presents the relationships between all land-use types in future land-use patterns and their capacity for ES provision (Table S3 in the Supplementary Materials, [47]).

### 2.4.2. Future Land-Use Patterns by Land-Use Strategies

Land-use strategies were spatially implemented as an aggregation of rearranged land-use patterns by agricultural land-management options. The CA module in GISCAME was used to simulate future land-use patterns according to spatially explicit rule-sets that govern how and where to apply future options [75]. The CA, which is a spatially discrete dynamic gridded model, updates states of cells, i.e., land-use types, in a defined area called the neighborhood based on locational

conditions of the cells [7,82]. The rule-sets were elaborated based on consulted information with the stakeholders regarding transition probabilities, neighboring land-use types, and environmental conditions. For instance, the consulted information included the probability (%) of land-use change from cereal (current state) to cereal intercropping (future state), neighboring land-use types (proximity effects), and environmental attributes (e.g., soil and slop conditions) as conversion conditions (Table S4 in the Supplementary Materials, [47]). When a land-use strategy is composed of cereal-dominant intercropping, maize-dominant intercropping, and mango agroforestry on legumes management options, a new land-use pattern can be generated by the simultaneous application of rule-sets of those management options through the CA (Figure 4).

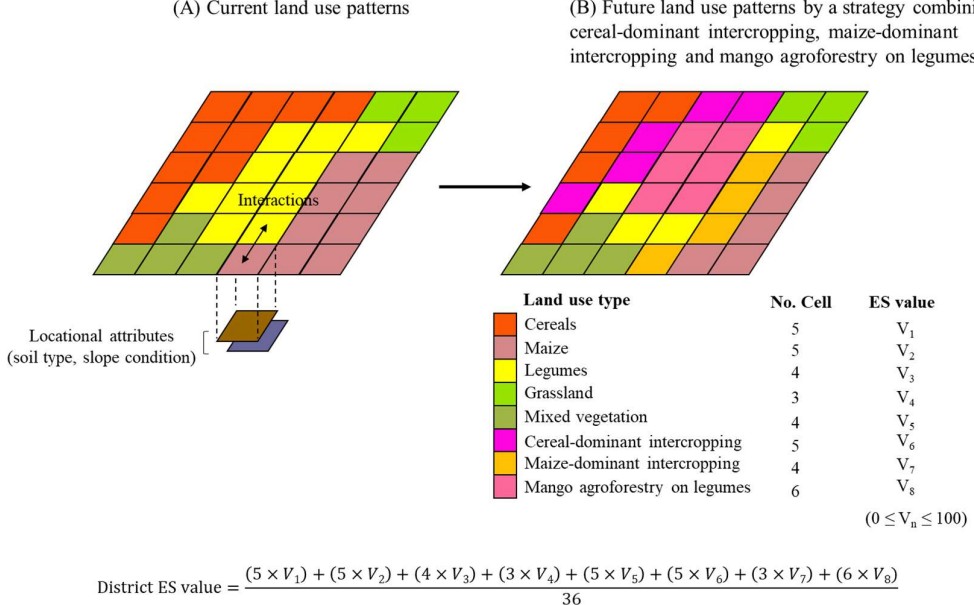

$$\text{District ES value} = \frac{(5 \times V_1) + (5 \times V_2) + (4 \times V_3) + (3 \times V_4) + (5 \times V_5) + (5 \times V_6) + (3 \times V_7) + (6 \times V_8)}{36}$$

**Figure 4.** Generation of future land-use patterns by a land-use strategy and resultant ecosystem service values at district level. According to transition rule-sets for land-use management options (interactions with neighboring cells and locational attributes), land-use patterns are changed from A to B. An ecosystem service value at district level is calculated as a mean value for the ecosystem service supplied by each cell of changed land-use patterns.

### 2.4.3. Identification of Ecosystem Services Values and Feasible Land-Use Strategies at District Level

The ES values of the whole district area were calculated as mean values for the ES provided by each land-use cell of rearranged land-use patterns according to future land-use strategies (Figure 4). In other words, the final assessment score indicated the mean capacity of the all land-use types in the district map to supply ES. The capacity of a district to provide ES according to the land-use strategies was expressed in a spider chart and an ES balance table. This representation of results allowed a visual comparison of the expected impact of land-use strategies, which were interpreted as trade-offs and synergies between ES. According to the ES values at district level, the most effective land-use strategies to provide each ES were determined. The best land-use strategies were identified that can supply more than three different ES with the highest ES potential.

The reflection of local perspectives is essential to identify feasible future strategies of a specific context [83]. Although a certain land-use strategy is assumed to be effective to enhance various ES based on the simulated results, the strategy might be unrealistic without the consent of stakeholders. In this sense, a structured stakeholder survey was conducted to investigate the feasibility of the best land-use strategies using a Likert-scale (from 0 = unrealistic to 5 = very likely) with the visualized simulation results. The mean and coefficient of variation of the feasibility level were used to identify

the land-use strategies with the highest feasibility to be adopted in the districts and the highest capacity to provide multiple ES.

## 3. Results

### 3.1. Ecosystem Services Values of Future Land-Use Strategies at District Level

As potential land-use alternatives, 75 future land-use strategies that consist of different management options (acronym) were evaluated (Table S5 in the Supplementary Materials). The application of the land-use strategies led to rearranged land-use patterns (examples in Figure 5a and Figure S1a in the Supplementary Materials), changes in spatial distribution of ES provision (examples in Figure 5b and Figure S1b in the Supplementary Materials), and altered ES provision at district level (examples in Figure 5c and Figure S1c in the Supplementary Materials). For example in Figure 5b, the amount of green areas (high capacity to provide food) in the map of strategy 13 was higher than in strategy 20. The spider chart and ES balance table (Figure 5c) also showed that the positive impact of strategy 16 on multiple ES was higher than strategy 20. Regarding the ES values in Bolgatanga as output of GISCAME for all 75 land-use strategies (Table S6 in the Supplementary Materials), most land-use strategies showed either no change or a positive impact on the provision of all ES, except for water provision. Specifically, land-use strategies that included cereal-dominant intercropping and legume-dominant intercropping (e.g., land-use strategies 1, 10, and 13) showed higher food provisioning levels than other strategies. On the contrary, land-use strategies that included mango agroforestry on cereals (e.g., land-use strategies 17, 19, 20, 23, 27, and 30) were considered less effective for providing food. The provision of fodder and construction materials was similarly increased by overall land-use strategies. Energy provision was higher in land-use strategies that included the windbreak as management option on cereals (e.g., land-use strategies 61–75). The provision of marketable products increased in land-use strategies that incorporated the combination of legume-dominant intercropping and soil or stone bunds on cereals (e.g., land-use strategies 46, 55, and 58), whereas the effect through land-use strategies that included agroforestry as management option (e.g., land-use strategies 20, 23, 30, 36, and 39) was lower than the effect by other strategies. Water provision was drastically decreased by all land-use strategies, especially land-use strategies that included mango agroforestry on cereals. According to the simulated results, strategy 13 (CI + MW + LI + GA + MxA) is one of the most effective strategies for increasing multiple ES, whereas strategy 20 (CM + MM+ LM + GA + MxA) is less effective than others.

With respect to ES values in Bongo (Table S7 in the Supplementary Materials), land-use strategies with cereal-dominant intercropping (e.g., land-use strategies 1, 2, and 11) were shown as more effective for food provision than others, while land-use strategies that included agroforestry on crops (e.g., land-use strategies 21–24, 33–39, and 45) proved to be less effective for increasing food provision. Unlike Bolgatanga, land-use strategies with leucaena (fodder tree) agroforestry on legumes (e.g., land-use strategies 33, 39, 42, 45, 48, 54, 57, and 60) proved to be effective for the increase in fodder provision. Energy provision was increased more through land-use strategies with a windbreak on cereals as a management option (e.g., land-use strategies 64–75) than through other strategies. The provision of construction materials was increased equally by most land-use strategies. Regarding the improved provision of marketable products, land-use strategies that included legume-dominant intercropping (e.g., land-use strategies 1, 4, and 46) presented to be more effective than others. Water provision was notably reduced by all land-use strategies dissimilar to other ES, and the negative effect was especially greater through land-use strategies with mango agroforestry on cereals (e.g., land-use strategies 19, 21, 24, 27, and 30). The enhancement of erosion control was more effective in land-use strategies with soil or stone bunds on cereals (e.g., land-use strategies 46–60). Simulated results showed that strategy 1 is one of the most effective strategies to enhance various ES, while strategy 36 is less effective than other strategies.

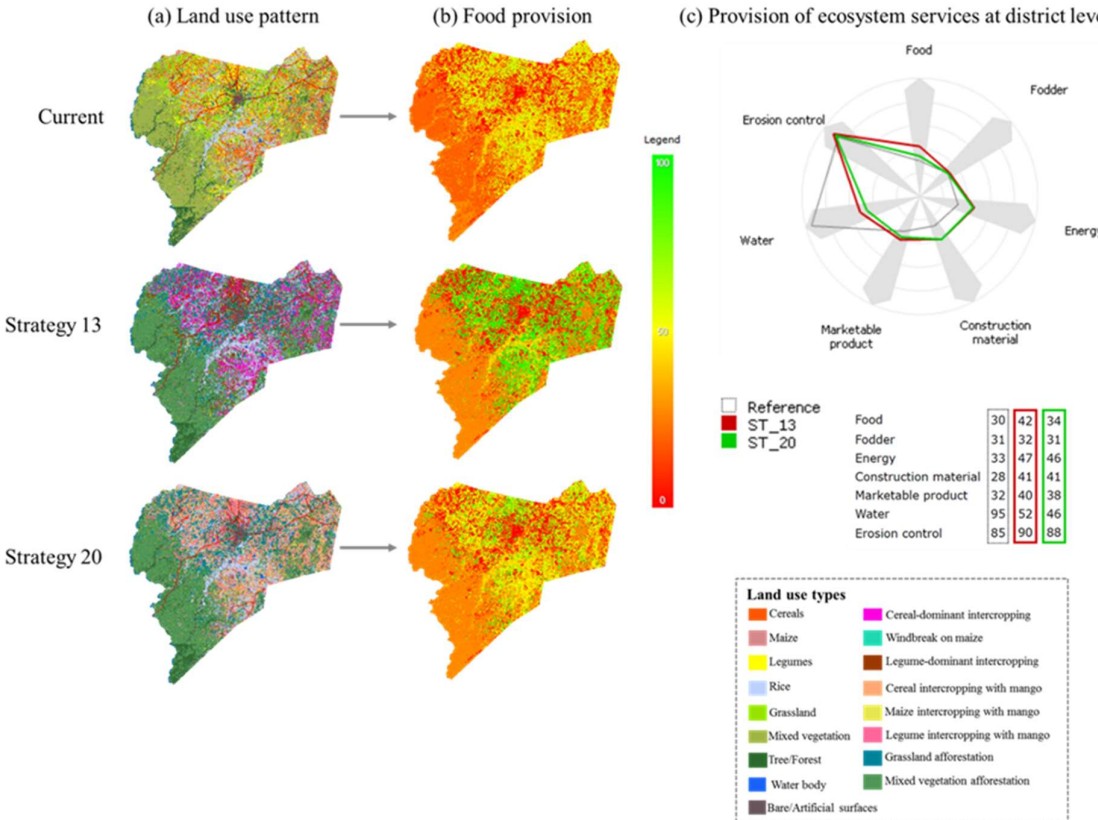

**Figure 5.** Potential impacts of land-use strategies on the land-use patterns and the provision of ecosystem services in Bolgatanga. The application of strategy 13 (ST_13) and strategy 20 (ST_20) results in rearranged land-use patterns (**a**). Provisioning maps for ecosystem services (e.g., food provision) show the impacts of the rearranged land-use patterns on the spatial distribution of ecosystem services (**b**). The spider chart and the ecosystem services balance table show changes in the provision of ecosystem services compared to the current provision of ecosystem services as reference (**c**). When these two strategies are compared, strategy 13 is more effective to enhance ecosystem services. The images were captured from GISCAME.

As the best land-use strategies that provide more than three different ES with the highest values (green color in Tables S6 and S7 in the Supplementary Materials), 14 land-use strategies in Bolgatanga and eight land-use strategies in Bongo were identified (yellow color in Tables S6 and S7 in the Supplementary Materials). The best land-use strategies in Bolgatanga were based on legume-dominant intercropping (LI) and soil conservation applied in cereals (CB, CW) as agricultural land-management options. Those strategies especially enhanced food provision more effectively than other ES. In Bongo, land-use strategies that contained soil or stone bunds on cereals (CB) and agroforestry in legumes (LM, LL) tended to be the best land-use strategies. They increased particularly the provision of food and marketable products. All the best strategies in Bolgatanga and Bongo led to a decrease in water provision as trade-off.

*3.2. Locally Recommendable Land-Use Strategies*

Among the best land-use strategies (14 strategies in Bolgatanga and eight strategies in Bongo), recommendable land-use strategies were determined in consideration of their feasibility in practice. The feasibility based on a stakeholder survey is presented in Table 2. In terms of Bolgatanga, most of the best land-use strategies were above the moderate level of feasibility (mean value ≥ 3). In particular, (I) a combination of crop intercropping on cereals, maize, and legumes, and afforestation on grassland and mixed (land-use strategy 1) and (II) a combination of windbreak on cereals, crop intercropping on maize and legumes, and afforestation on grassland and mixed vegetation (land-use strategy

61, visualized impact on ES provision and land-use patterns in Figure 6) presented slightly higher mean values and lower variation. Therefore, these two land-use strategies can be considered as locally recommendable land-use strategies that have the potential to enhance multiple ES with high feasibility to be implemented in the local context. On the other hand, land-use strategy 64, which consisted of windbreak on cereals, maize intercropping with mango, legume-dominant intercropping, and afforestation on grassland and mixed vegetation, was perceived as being less feasible to be adopted.

**Table 2.** The feasibility of best land-use strategies in Bolgatanga and Bongo. Mean values and the coefficient of variation (CV) of a Likert-scale survey result (from 0 = unrealistic to 5 = very likely) are used for the identification of most feasible land-use strategies.

| Bolgatanga | | | |
|---|---|---|---|
| | | **Feasibility** | |
| **Nº** | **Land-use strategy** | **Mean** | **CV** |
| 1 | CI + MI + LI + GA + MxA | 3.78 | 0.12 |
| 10 | CI + MB + LI + GA + MxA | 4 | 0.25 |
| 13 | CI + MW + LI + GA + MxA | 3.78 | 0.26 |
| 46 | CB + MI + LI + GA + MxA | 3.56 | 0.25 |
| 55 | CB + MB + LI + GA + MxA | 3.67 | 0.14 |
| 57 | CB + MB + LL + GA +MxA | 3.78 | 0.22 |
| 58 | CB + MW + LI + GA + MxA | 3.22 | 0.21 |
| 61 | CW + MI + LI + GA + MxA | 4.22 | 0.16 |
| 63 | CW + MI + LL + GA + MxA | 3.56 | 0.25 |
| 64 | CW + MM + LI + GA +MxA | 2.56 | 0.21 |
| 67 | CW + ML +LI +GA + MxA | 3.44 | 0.29 |
| 70 | CW + MB + LI + GA + MxA | 3.67 | 0.19 |
| 73 | CW + MW + LI + GA + MxA | 3.22 | 0.37 |
| 75 | CW + MW + LL + GA +MxA | 3.11 | 0.30 |
| Bongo | | | |
| | | **Feasibility** | |
| **Nº** | **Land-use strategy** | **Mean** | **CV** |
| 1 | CI + MI + LI + GA + MxA | 3.78 | 0.18 |
| 2 | CI + MI + LM + GA + MxA | 4.10 | 0.18 |
| 46 | CB + MI + LI + GA + MxA | 4.11 | 0.15 |
| 47 | CB + MI + LM + GA + MxA | 3.78 | 0.18 |
| 48 | CB + MI + LL + GA + MxA | 3.56 | 0.20 |
| 54 | CB + ML + LL + GA + MxA | 3.78 | 0.26 |
| 57 | CB + MB + LL + GA +MxA | 3.89 | 0.15 |
| 60 | CB + MW + LL + GA +MxA | 3.44 | 0.29 |

All best land-use strategies in Bongo that provide the highest ES potential for multiple ES showed also higher feasibility than the moderate level (mean value ≥ 3). Especially, (I) a combination of soil or stone bunds on cereals, crop intercropping on maize and legumes, and afforestation on grassland and mixed vegetation (land-use strategy 46, visualized impact on ES provision and land-use patterns in Figure 6) and (II) a combination of crop intercropping on cereals and maize, legume intercropping with mango, and afforestation on grassland and mixed vegetation (land-use strategy 2) presented

slightly higher feasibility than other land-use strategies considering their mean values and coefficient of variation. Thus, they can be regarded as locally recommendable land-use strategies in Bongo. A combination of soil or stone bunds on cereals, windbreak on maize, legume intercropping with leucaena, and afforestation on grassland and mixed vegetation (land-use strategy 60), on the contrary, was regarded as a less feasible strategy.

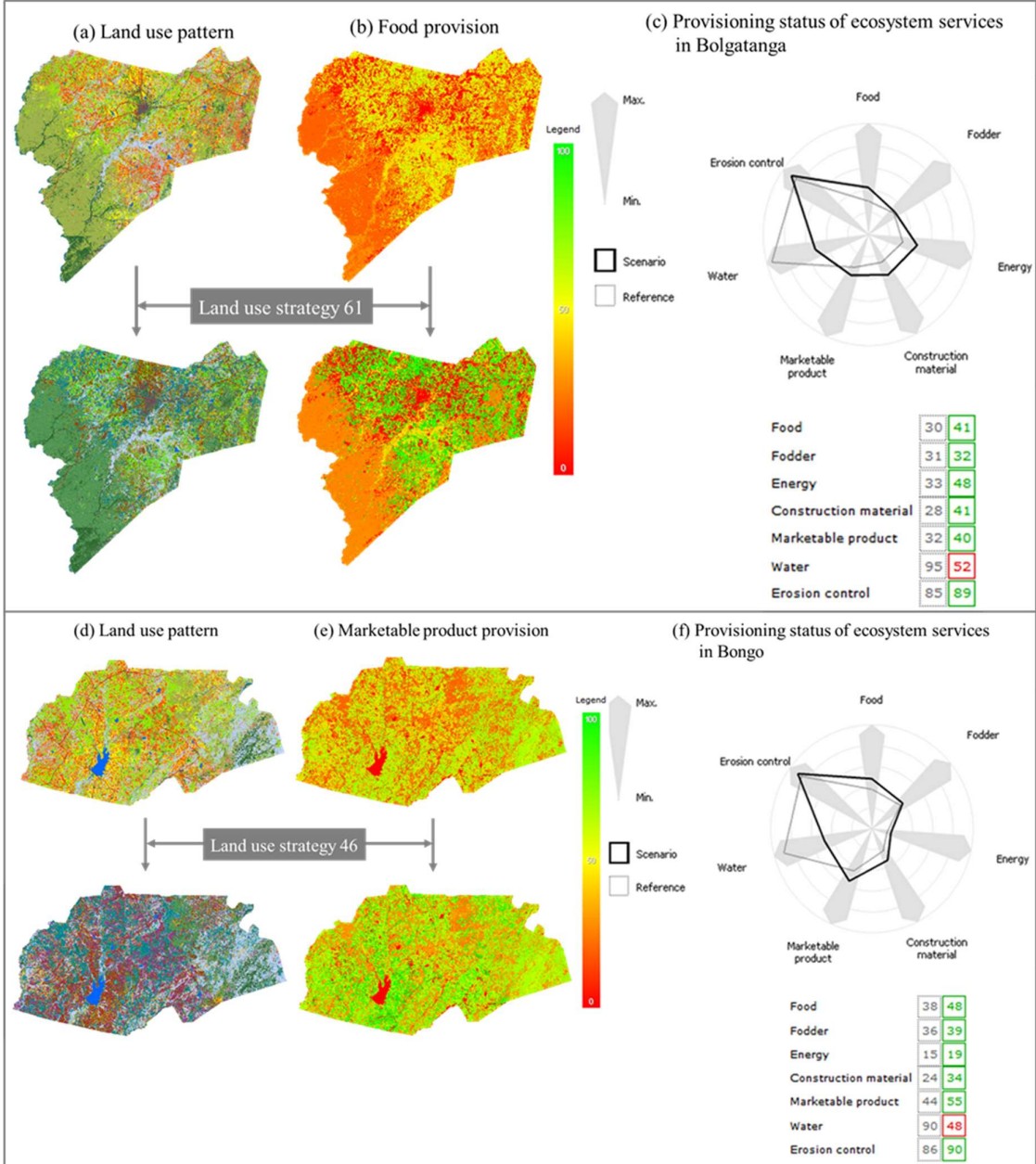

**Figure 6.** Future land-use pattern and changes in the provision of ecosystem services according to locally recommendable land-use strategies in Bolgatanga (top) and Bongo (bottom). The application of land-use strategies generates future land-use pattern (**a**,**d**). The spatial distribution of ecosystem services (e.g., food and marketable products provision) is influenced by the newly generated land-use pattern (**b**,**e**). The spider chart and the table present the changes in the provision of overall ecosystem services at district level compared to the current status as reference values (**c**,**f**). Green numbers indicate an increase in the provision of ecosystem services, and a red number signifies a decrease in the provision of ecosystem services. The images were captured from GISCAME.

## 4. Discussion and Outlook

### 4.1. Discussion of the Findings

The quantified ES values enabled an understanding of the potential impacts of land-use strategies associated with district characteristics and local perception. For instance, since cereals were the main staple crops in both districts (Table S1 in the Supplementary Materials), food provision was mainly influenced by the management option applied for cereal fields. Different perceptions existed on the capacity of the cereal-related management options to provide food (Table S3 in the Supplementary Materials). For example, land-use strategies with agroforestry on cereals were considered to provide less food than land-use strategies including cereal-dominant intercropping. In terms of the provision of marketable products, legume-dominant intercropping showed the highest capacity to provide the ES (Table S3 in the Supplementary Materials) and substantially contributed to the effectiveness of land-use strategies in both districts. Different perceptions explain dissimilarities between the two districts regarding the ES provision of land-use strategies. For example, management options with leucaena agroforestry for fodder provision and soil or stone bunds for erosion control were considered to be more effective according to the opinion of the stakeholders in Bongo. The stakeholders in Bolgatanga perceived cereal-dominant intercropping management options as more effective for providing those ES. This can be explained by the different experiences of stakeholders in the fields and by more economic oriented preferences for land-use practices. Bolgatanga as regional capital hosts the main markets and has a higher purchasing power than Bongo. Therefore, "bestsellers" such as cereals have been chosen to ensure the economic income of farmers. In addition, the decentralized program in Ghana that enabled individual decision-making system for each district [53] also influences such differences between the districts. Malinga et al. [84] and Mensah et al. [85] also found out with case studies in South Africa that ES provided by a certain landscape and land-use system, and their usage in practice can be differently perceived depending on the socio-economic status (age, gender, income, etc.), knowledge, and experiences of the stakeholder group.

All land-use strategies presented a remarkable decrease in water provision as a trade-off to the increase in other ES (Tables S6 and S7 in the Supplementary Materials). Since water provision here is defined as the amount of surface water directly used by people, land-use types with a high-water demand lead to a negative impact on the ES [47]. For instance, intercropping and agroforestry practices were considered to highly increase surface water demand due to the varied water requirements of different intercropped species in both districts (Table S3 in the Supplementary Materials). Thus, land-use strategies as combinations of various management options that have a higher water demand could potentially amplify such a negative impact on the provision of water at district level. Previous studies also showed that combined management options (e.g., intercropping) led to an increase in the total surface water demand and water stress, thereby reducing water yield [86,87]. However, there has been a debate about the effect of land-use practices on water yield, and this is closely related to spatial scales. Specifically, in Africa, the role of trees and forests is often focused as water consumers and competitors for other water uses at local level, while they provide water to the atmosphere and contribute to precipitation development at regional and global level [88,89]. This interaction between land-use types and the provisioning level of atmospheric moisture should be especially considered in semi-arid West Africa. In this assessment, all selected ES were equally weighted as done in other existing ES studies [90–92]. The application of different weight values to ES, which allows us to reflect preferred or prioritized ES from the stakeholder perspective could present more realistic results regarding trade-offs and synergies between ES. When different importance of ES is considered, specific land-use strategies can be recommended. For instance, if the stakeholders regard that food provision is most crucial in this area, strategy 1 can be more recommendable than strategy 61 in Bolgatanga, and strategy 2 can be chosen as a recommendable strategy rather than strategy 46 in Bongo due to their more effective capacity to provide food. In this light, future land-use strategies for supporting the

efficient use of limited agricultural land focusing on the provision of more important ES in the local context could be also identified.

*4.2. Methodological Discussion*

This study integrated a participatory method and spatially explicit simulation modeling as a transdisciplinary approach to apply the ES concept for assessing future land-use strategies. Table 3 presents advantages and challenges of the applied methods. A participatory method through stakeholder surveys allowed us to reflect local preferences and characteristics in assessing the potential impacts on multiple ES provision. Land-use strategies based on local perspectives allowed the development of more acceptable and feasible future land-use alternatives. However, stakeholder perspective-based data might cause a reliability issue and the ignorance of important environmental aspects. In addition, the involvement of only a certain group in the assessment process has a limitation to address conflicting objectives.

Regarding spatially explicit simulation modeling, GISCAME as an assessment platform runs with simplified data reflecting locally relevant details, thereby easily testing various future alternatives. The visualization of ES provision according to the simulated results can improve the understanding of potential impacts of future decisions (trade-offs and synergies between ES). Quantified and visualized results allowed a better feedback from stakeholders. Furthermore, such simulation approaches can also improve the communication between different land-use actors, which helps to establish shared understandings and visions on future actions. On the other hand, modeling always deals with an abstract of the complex environment. Various direct and indirect factors influence agricultural systems, but the applied modeling approach simplified the environmental factors that influence land-use decisions of the stakeholders. Another challenge of this local ES assessment is related to the transferability to other regions or different spatial scales. The findings are based on the context-specific as well as scale-dependent empirical data, which were generated by the stakeholder involvement. However, the applied assessment framework, which presented the stepwise process of collecting ES and land-use-related local knowledge and integrating the local knowledge into land-use modeling, can be used in other contexts and regions. GISCAME, which was used for spatially explicit simulation in this framework has been already applied in Germany, Chile, Ecuador, and Brazil [64,75,93]. Future research in the context of this study can be directed towards scalar interactions of land-use systems, i.e., land-use decisions at district level influence the land-use conditions at farm level and land-management policies at regional level. Depending on the spatial scales, the stakeholder group should be adapted, since other stakeholders might be more relevant on the regional and national level.

**Table 3.** Advantages and challenges of applied participatory and simulation modeling approaches.

| | Advantage | Challenge |
|---|---|---|
| Participatory method | • Local preferences and characteristics were reflected in identifying the relationships between future land-use strategies and ES provision.<br>• ES and indicators were identified relevant to actual land-use activities in the local context: the multifunctionality of land-use systems can be considered [94,95].<br>• Acceptable and feasible land-use strategies were generated based on agricultural land-management options from a local perspective: this can complement existing statistical and biophysical data-based scenario assessments in West Africa (e.g., [22,45,46,96]). | • Reliability of results can be criticized due to the subjective data based on the perspectives of the stakeholders.<br>• Important environmental aspects may not have been considered by the stakeholders (e.g., impact of land-use systems on climate regulation service).<br>• Only a specific stakeholder group was involved: potential conflicts and trade-offs between the interests of different actors were not considered [97,98]. |
| Spatially explicit simulation modeling | • It can incorporate stakeholders' perspectives vis-à-vis the spatial peculiarity of ES provision, whose distribution and values are dependent on land-use patterns [8,99,100].<br>• It has potential to be used as a transdisciplinary planning approach that integrates a participatory method and ES mapping, especially in the West African context, where locally adapted methodological frameworks are still limited [18].<br>• GISCAME runs with simplified data reflecting locally relevant details rather than requiring extensive and big-data, which allows easier integration with various types of local data and transformation of the modeled results into decision-making relevant information<br>• The visualization of ES provision can improve the understanding of potential impacts of future decisions and can support land-use decision-making and planning as an ex-ante assessment of future land-use alternatives [101,102].<br>• Quantified and visualized results allow stakeholders to compare different alternatives and to be actively involved in a decision process.<br>• The approach can be used as feedback mechanism and also as a communication tool between different stakeholder groups [103]. | • A simplification of the complex environment was needed for modelling [104,105]: dynamics of interactions between future land-use decisions and ES provision were limited.<br>• Agricultural conditions are greatly influenced by various direct and indirect factors such as the use of fertilizers labor availability, subsidy programs, and market situation [106,107], which were not included due to the increasing complexity and the lack of adequate data.<br>• The transferability of results to other regions or different spatial scales is limited because the applied data contains stakeholder-specific knowledge [108,109].<br>• The analysis was conducted at district level, which is nested between the field and national level [110]. However, the scalar interactions were not considered due to the modeling complexity and the lack of regional data for multi-scale assessments. |

*4.3. Future Directions of Using the Ecosystem Service Concept for Land-Use Planning in West Africa*

The integration of the social and the ecological systems is essential for land-use decision making [3,32,101,111,112] and the involvement of stakeholders improves the understanding of such linkages [109]. The transdisciplinary concept of ES could serve as a bridge between the social and ecological system and different actors. Therefore, the ES concept has the potential to contribute to participatory land-use planning [35,113]. However, the implementation of the ES concept in actual planning and decision making is still in the initial stage and existing approaches to make use of ES values need to be further tested in practice [114], especially in West Africa. Previous studies addressed the challenges to apply the ES concept in spatial planning in West Africa, which are related to the lack of awareness and common understanding of the ES concept, low public participation, and the lack of tools and approaches to support practical implementation of land-use strategies [18,26]. Since local people in West Africa are still unacquainted with the ES concept and related scientific terms, we used "benefit of agricultural land-use activities" instead of "ES" during discussion and surveys with the local stakeholders. ES indicators were also determined to reflect their consumptive patterns of the benefits, which the multifunctionality of agricultural land was considered from the local perspective.

We adopted an assessment framework that used qualitative and semi-quantitative data as simulation input and evaluation of ordinal scale for identifying locally recommendable strategies. This approach is useful in applying the ES concept in the West African context, since it can serve as a preliminary basis for decision-making through presenting changes in ES provision depending on probable future decisions of the local stakeholders. Such consideration of local perspectives allows the better public understanding of the ES concept [35,113] and increases the acceptance of the ES assessment results. The involvement of various agricultural related actors (e.g., farmers, NGOs, agribusinesses, other governmental bodies, and experts) in the feedback process can increase the validity of the findings and further support consensus building thereby encouraging collective actions [113].

In order to make better use of ES assessment for land-use planning in West Africa, it should be investigated how ES information can be operationalized in a specific policy context [115]. Regarding land-use planning, the spatial distribution and peculiarity of ES in a certain area is a key information since stakeholders and decision-makers are more interested to know where to implement planning as a spatial solution [113]. Such information can be more applicable with practical knowledge respecting how and when the information and tested approaches can actually support planning practice [114]. In Ghana, land-use planning has been criticized for focusing mostly on managing physical growth and developing urban areas, despite the fact that the majority of the land still needs to be used for food and natural resources. Besides, the ES concept has been so far rarely emphasized in any Ghanaian spatial development schemes [18]. Thus, there should be further research concerning which ES-relevant information is required by planners and decision-makers and how to establish a new standard or criteria of ES plans coordinated with existing decision-making structures [114].

## 5. Conclusions

This study suggested an assessment framework to support future land-use planning for agricultural land through integration of local knowledge into spatially explicit ES simulation modeling. Considering that existing studies for assessing the impact of land-use systems in West Africa, which did not consider either local perspectives or the spatial peculiarity of ES, the applied approach in this study can be a novel attempt to connect narratives of stakeholders and explicit approaches. Especially, the development of land-use strategies based on stakeholder perspectives allowed identification of more accountable alternatives for effective ES-based adaptation in the local context. Converted local knowledge and perception to model input for spatially explicit simulation allowed to understand the interrelationships between future land-use decisions by stakeholders, changes in land-use patterns, and their consequent impact on ES provision. The results reflecting different local perceptions on the land-use systems presented that different land-use strategies were regarded as effective and feasible in the two adjacent districts despite their similar land use and environmental conditions. This implies that local knowledge and characteristics such as the multifunctionality of land-use systems and locally preferred land-use activities, which could be influenced by socio-economic factors and decision-making process of districts are important in identifying effective future strategies for improving locally relevant ES. The quantified and visualized impacts of land-use strategies facilitated the communication with the local stakeholders for obtaining their feedback. This shows the potential of a modeling approach to contribute to elaborating locally tailored land-use schemes as a transdisciplinary way. As a stakeholder-based simulation modeling, there are some weaknesses to contemplate regarding the simplification of complex human-nature systems, transferability of results to other regions or spatial scales and limitations in considering various socio-economic aspects due to the lack of data. In addition, the involvement of various actors in assessment and feedback processes should be also considered. However, the suggested modeling approach gives an insight into how to design decision-supporting frameworks for future land-use planning from the transdisciplinary perspective, which reflects the interaction between land-use stakeholders and their surroundings through an integration of different methods.

**Supplementary Materials:** The following are available online at http://www.mdpi.com/2073-445X/9/10/379/s1, Figure S1: Potential impacts of land-use strategies on the land-use patterns and the provision of ecosystem

services in Bongo. Strategy 1 (ST_1) and strategy 36 (ST_36) lead to rearranged land-use patterns (a). The spatial distribution of ecosystem services (e.g., food provision) is changed according to the strategies (b). Impacts on the provision of ecosystem services at district level compared to the current status as reference are expressed in the spider chart and the ecosystem services balance table (c). When these two strategies are compared, strategy 1 is more effective to enhance ecosystem services. The images were captured from GISCAME. Table S1: The percentage of the area occupied by each land-use type and their descriptions [27]. Table S2. Agricultural land-management options and their description [47]. Table S3. Ecosystem services assessment matrix to display the capacity of current land-use types and agricultural land-management options to provide ecosystem services in Bolgatanga and Bongo [47]. The values are presented within a scale from 0 (lowest level of provision) to 100 (highest level of provision). Table S4. Transition probability-based application conditions for land-management options [47]. Table S5. Applied future land-use strategy. Table S6. Ecosystem service values provided by land-use strategies in Bolgatanga. Ecosystem service values based on the current land-use pattern are used as reference values (R), in blue color. The highest value of each ecosystem service is expressed as green color (the provision of construction materials is excluded as it is equally increased by all land-use strategies). The best land-use strategies that have the potential to provide more than three different ES with the highest values are expressed as yellow color. Table S7. Ecosystem service values provided by land-use strategies in Bongo. Ecosystem service values based on the current land-use pattern are used as reference values (R), in blue color. The highest value of each ecosystem service is expressed as green color (the provision of construction materials is excluded as it is equally increased by all land-use strategies). The best land-use strategies that have the potential to provide more than three different ES with the highest values are expressed as yellow color.

**Author Contributions:** Conceptualization, H.K.; Data curation, H.K.; Formal analysis, H.K.; Funding acquisition, C.F.; Investigation, H.K.; Methodology, H.K.; Project administration, C.F.; Resources, H.K.; Software, C.F.; Supervision, C.F.; Validation, H.K.; Visualization, H.K., J.K.; Writing-original draft, H.K.; Writing-review & editing, H.K., J.K. All authors have read and agreed to the published version of the manuscript.

**Funding:** The work was conducted within the West African Science Service Center on Climate Change and Adapted Land Use (WASCAL) project, funded by the German Federal Ministry of Education and Research (BMBF) under grant numbers at the Center for Development Research (ZEF) [00100218] and the Karlsruhe Institute of Technology (KIT) [5260.0109.3288].

**Acknowledgments:** The authors are grateful to the agricultural extension officers of Ministry of Food and Agriculture in Bolgatanga and Bongo districts who participated in data collection as key stakeholders, and the directors of the districts who helped to organize meetings with the officers. The authors also acknowledge the financial support within the funding program Open Access Publishing by the German Research Foundation (DFG).

**Conflicts of Interest:** The authors declare no conflict of interest.

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
