# Peer review of "Integrating Ecosystem Services into Land-Use Modeling to Assess the Effects of Future Land-Use Strategies in Northern Ghana"

_land, doi:10.3390/land9100379_

Round 1

Reviewer 1 Report

Thank you very much for your thorough revision of the manuscript. It is now considerably improved, methods and materials are much clearer and easier to follow. The background on stakeholder selection and the participatory approach is also very useful.

Introduction:

line 42- 54 List of case studies - one sentence, one study. This can be rephrased and restructured to make for easier reading

Materials and Methods

line 255 - 257 What criteria did the local stakeholders use to assess feasibility of the different strategies?

line 264 that were assessed

Results:

Consider moving table 2 to the appendix. The information is already contained in Figure 2. 

Author Response

Review #1

Thank you very much for your thorough revision of the manuscript. It is now considerably improved, methods and materials are much clearer and easier to follow. The background on stakeholder selection and the participatory approach is also very useful.

Introduction:

line 42- 54 List of case studies - one sentence, one study. This can be rephrased and restructured to make for easier reading

Thanks for your comment. We restructured this part according to the similarity of the cited studies (L42-49):

“Taking case studies of Europe, Asia and America as an example, the impact of past and current land use and land cover changes on the provision of specific ES (e.g. fresh water provision and air quality) was analyzed in order to give an insight into relevant land use schemes and policies [10,11]. Different future pathways according to social-ecological drivers and their potential implications to the ES status were explored and discussed based on the experts and stakeholders’ opinions [12, 13]. In addition, changes in multiple ES provision depending on the future land use and landscape patterns were quantified for identifying optimal future options considering ES trade-offs [14-16].”

Materials and Methods

line 255 - 257 What criteria did the local stakeholders use to assess feasibility of the different strategies?

Thanks for your comment. In this section, we described the overall assessment process according to the presented framework in Figure 3. We elaborated how to identify the feasibility of the different strategies from the local perspectives in the section 2.4.3 (L309-313):

“In this sense, a structured stakeholder survey was conducted to investigate the feasibility of the best land use strategies using a Likert-scale (from 0= unrealistic to 5= very likely) with the visualized simulation results. The mean and coefficient of variation of the feasibility level were used to identify the land use strategies with the highest feasibility to be adopted in the districts and the highest capacity to provide multiple ES.”

line 264 that were assessed

Thanks. We revised it as requested.

Results:

Consider moving table 2 to the appendix. The information is already contained in Figure 2.

Thanks for your comment. We moved the table to the Supplementary Materials section according to your suggestion (Tab. S5).

Reviewer 2 Report

In this resubmitted version of the paper, the authors more clearly explained the relationship of the present studies with the ones previously published and provided a more elaborated discussion on advantages and limitations on the used method.

Before   publication, I suggest that a final proofread is carried out to fix some minor errors/inaccuracies, e.g.

Line 182 “as a field expert” change to “as field experts”  

line 463-464 “In this assessment, all selected ES were equally weighted as existing ES studies [90-92]” better: “In this assessment, all selected ES were equally weighted as done in other existing ES studies [90-92]”

Author Response

Reviewer #2

In this resubmitted version of the paper, the authors more clearly explained the relationship of the present studies with the ones previously published and provided a more elaborated discussion on advantages and limitations on the used method.

Before publication, I suggest that a final proofread is carried out to fix some minor errors/inaccuracies, e.g.

Line 182 “as a field expert” change to “as field experts”

Thanks for your comment. We changed it to “field experts” as requested.

line 463-464 “In this assessment, all selected ES were equally weighted as existing ES studies [90-92]” better: “In this assessment, all selected ES were equally weighted as done in other existing ES studies [90-92]”

Thanks for your comment. We changed it to “…as done in other existing ES studies” as requested.

Additionally, we revised some errors such as:

L357: The best land use strategies -> As the best land use strategies

L441: a high water demand -> a high-water demand

L 561: consider -> reflects
